# Analysis of Factors Determining Patient Survival after Receiving Free-Flap Reconstruction at a Single Center—A Retrospective Cohort Study

**DOI:** 10.3390/diagnostics12112877

**Published:** 2022-11-21

**Authors:** Nicholas Moellhoff, Sara Taha, Nikolaus Wachtel, Maximilian Hirschmann, Marc Hellweg, Riccardo E. Giunta, Denis Ehrl

**Affiliations:** Division of Hand, Plastic and Aesthetic Surgery, University Hospital, LMU Munich, Marchioninistraße 15, 81377 Munich, Germany

**Keywords:** multimorbidity, free flap reconstruction, gracilis free flap, latissimus dorsi free flap, microcirculation

## Abstract

Background: Microsurgical tissue transfer revolutionized reconstructive surgery after extensive trauma, oncological resections, and severe infections. Complex soft tissue reconstructions are increasingly performed in multimorbid and elderly patients. Therefore, it is crucial to investigate whether these patients benefit from these complex procedures. Objective: To evaluate the outcome for multimorbid patients who underwent microsurgical soft tissue reconstruction and to identify potential risk factors that may increase mortality. Methods: This single-center study retrospectively analyzed prospectively collected data of patients receiving free gracilis (GM) or latissimus dorsi muscle (LDM) flap reconstruction between September 2017 and December 2021. Cases were divided into two groups (dead vs. alive), depending on patient survival. Patient demographics, comorbidities and medication, perioperative details, free flap outcome, as well as microcirculation were determined. Results: A total of 151 flaps (LDM, *n* = 67; GM, *n* = 84) performed in 147 patients with a mean age of 61.15 ± 17.5 (range 19–94) years were included. A total of 33 patients (22.45%) passed away during the study period. Deceased patients were significantly older (Alive: 58.28 ± 17.91 vs. Dead: 71.39 ± 11.13; *p* = 0.001), were hospitalized significantly longer (Alive: 29.66 ± 26.97 vs. Dead: 36.88 ± 15.04 days; *p* = 0.046) and suffered from cardiovascular (Alive: 36.40% vs. Dead: 66.70%; *p* = 0.002) and metabolic diseases (Alive: 33.90% vs. Dead: 54.50%; *p* = 0.031) more frequently, which corresponded to a significantly higher ASA Score (*p* = 0.004). Revision rates (Alive: 11.00% vs. Dead: 18.20%; *p* = 0.371) and flap loss (Alive: 3.39% vs. Dead: 12.12%; *p* = 0.069) were higher in patients that died by the end of the study period. Conclusions: Free flap transfer is safe and effective, even in multimorbid patients. However, patient age, comorbidities, preoperative ASA status, and medication significantly impact postoperative patient survival in the short- and mid-term and must, therefore, be taken into account in preoperative decision-making and informed consent.

## 1. Introduction

Complex microsurgical reconstruction with free tissue transfer is increasingly performed in multimorbid and elderly patients, since the introduction of free microsurgical tissue transfer in the 1970s [1]. This technique leads to significantly improved surgical reconstruction after extensive trauma, oncological resections, and after severe infections. Moreover, in the field of oncological surgery, a more radical tumor resection can be carried out as complex and large defects may be covered by using free flaps [2,3,4,5,6,7,8,9,10,11,12]. The establishment of better microsurgical equipment, surgical techniques, operative techniques, and advanced monitoring systems has led to improved flap outcome, with a flap survival as high as 95–98% [2,3,4]. However, postoperative complications seem to become more relevant in multimorbid and elderly patients. Nevertheless, alternative treatment options such as major amputation show high mortality rates and reduced life quality in this patient cohort [3,4,5,6,7]. Additionally, the positive impact after microsurgical surgeries such as progress in rehabilitation and fast track concepts has an impact [13,14,15]. In clinical practice at a high-volume tertiary care hospital, the patient collective requiring free flap reconstruction shows a mix of complex diseases. At the same time, free flap surgery poses significant stress to the patient, with extensive operation and anesthesia time, a potential temperature deregulation due to largely exposed body areas, or hemodynamic instability requiring compensation by blood transfusions, catecholamines, and crystalloid or colloid infusions [4,5,6,16]. Multimorbidity certainly increases the perioperative risk, and patient mortality during or after surgery must be included into the evaluation of different treatment options. While success rates of free flap surgery are high, success is often defined as flap outcome in the short-term follow-up period. Irrespective of flap survival, however, the question needs to be posed to what extent multimorbidity has an impact on patient rather than flap survival [7]. Thus, the purpose of this single-center retrospective cohort study was to evaluate the outcome for multimorbid patients who underwent microsurgical soft tissue reconstruction using either a free gracilis (GM) or latissimus dorsi muscle (LDM) flap and to identify potential risk factors that may increase patient mortality.

## 2. Materials and Methods

### 2.1. Study Design

This single-center study retrospectively analyzed prospectively collected data of patients receiving free muscle flap reconstruction between September 2017 and December 2021. Cases were divided into two groups (dead vs. alive), depending on patient survival. Ethical approval was provided by the local ethics committee (21-04753, LMU Munich, Germany). The study was performed according to the principles of the Declaration of Helsinki 1996 and good clinical practice.

### 2.2. Patients

All patients receiving free GM or LDM flaps for any defect reconstruction within the given study period were included into analysis. Minors and pregnant patients were excluded; apart from that, no distinct exclusion criteria were defined. Follow-up was continued postoperatively until 1 February 2022, or was terminated upon patient death.

### 2.3. Data Collection and Outcome Parameters

Source data, including medical files, surgery reports, premedication records, and doctor’s letters, were screened for demographics, patient characteristics, perioperative details, flap survival, flap revision (arterial, venous thrombosis, and hematoma), and surgical and medical complications. The patients’ physical status was classified with the ASA score (American Society of Anesthesiologists). Indication for free flap surgery was categorized into three groups, including tumor resection, trauma, or wound healing disorder. Comorbidities were categorized into the following groups: cardiological diseases, vascular diseases, metabolic diseases, tumors, internal diseases, addiction to substances, neurological diseases, musculoskeletal diseases, and infectious diseases (see Table 1 for overview and examples for each category). Furthermore, patient medication prior to surgery was grouped into various categories for comparability purposes (see Table 2 for overview and examples for each category).

Patient survival was defined by the time between the date of free flap surgery and the date of death.

In addition, data on microcirculation were gathered for both flap entities utilizing laser-doppler flowmetry and tissue-spectrometry (O2C, LEA Medizintechnik, Gießen, Germany) upon availability of the measuring device. A previously described protocol was utilized for continuous measurement of venous-capillary microvascular flow (flow), hemoglobin oxygenation (SO_2_), and the relative amount of hemoglobin (rHb) [17,18,19,20]. In brief, measurements were performed continuously over a time period of 72 h post-anastomosis. They were interrupted only for patient transportation, probe dislocation, or to correct signal interferences during wound dressing changes. Only measurements of viable free flaps with an uneventful postoperative course (no major complications, revision surgery, or flap loss) were included in O2C data analysis to measure physiologic microcirculation in GM and LDM flaps.

### 2.4. Statistical Analysis

Data are presented as means with standard deviation, or as absolute and relative frequencies unless stated otherwise. Data were analyzed for normal distribution using the Shapiro–Wilk test and normal Q–Q plots. Student’s independent *t*-test was utilized to determine group differences on continuous dependent variables for normally distributed data. For non-normally distributed data on a continuous or ordinal dependent variable, the Mann–Whitney U test was applied. The chi-square or Fisher’s exact test was utilized to detect differences on dichotomous dependent variables. All calculations were performed using SPSS Statistics 28 (IBM, Armonk, NY, USA). Even if a non-parametric test was utilized, the mean value and the standard deviation were provided for better readability. Results were considered statistically significant at a probability level of ≤0.05. Mean values of flow, SO_2_, and rHb were extracted using the O2CevaTime Software (Version No. 28.3, LEA Medizintechnik, Gießen, Germany) according to a previously described protocol [14,15]. Kaplan–Meier survival analysis compared patient survival after free flap surgery.

## 3. Results

A total of 151 flaps (LDM, *n* = 67; GM, *n* = 84) performed in 147 patients with a mean age of 61.15 ± 17.5 (range 19–94) years were included in this clinical study. Patient demographics and perioperative details are summarized in Table 3. A total of 33 (22.45%) patients passed away during the study period. The mean survival time in patients that passed-away postoperatively was 403.30 ± 435.40 days. Kaplan–Meier survival analysis is presented in Figure 1. The mean follow-up time was 678.09 ± 473.56 days (range 0–1601 days).

### 3.1. Factors Affecting Patient Survival

When comparing free flaps of dead and alive patients, no statistically significant differences were found regarding gender, defect etiology, defect location, flap type, time of surgery, or time of flap ischemia. Interestingly, groups differed significantly with regard to age, with patients that had passed away at the end of the study period being significantly older, compared to those alive (Alive: 58.28 ± 17.91 vs. Dead: 71.39 ± 11.13; *p* = 0.001). In addition, the postoperative hospitalization was significantly higher in this population (Alive: 29.66 ± 26.97 vs. Dead: 36.88 ± 15.04 days; *p* = 0.046). Revision rates were higher in the group of patients that passed away in the postoperative follow-up period; however, no statistically significant difference was found with *p* = 0.371. Similarly, flap loss was higher in this group of patients, with rates being as high as 12.1%, as compared to 3.4%, with *p* = 0.069.

Comorbidities are tabulated within Table 4. Overall, the data show that patients who died in the postoperative period after free flap transfer suffered siginificantly more frequently from cardiovascular (Alive: 36.40% vs. Dead: 66.70%; *p* = 0.002) and metabolic diseases (Alive: 33.90% vs. Dead: 54.50%; *p* = 0.031). To add to that, the ASA score was significantly higher in this group (*p* = 0.004).

Detailed information about patients’ permanent preoperative medication is provided within Table 5. Once more, differences between groups were apparent, with patients who had passed away after free flap transfer having taken significantly more often anticoagulant/antiplatelet medication (Alive: 58.50% vs. Dead: 81.80%; *p* = 0.014), antidiabetic medication (Alive: 12.70% vs. Dead: 30.30%; *p* = 0.016), as well as medication for other metabolic diseases (Alive: 32.20% vs. Dead: 63.60%; *p* = 0.001).

### 3.2. Analysis of Microcirculation

Detailed analysis of microcirculation in viable free flaps showed comparable trends over time for flow, SO_2_, and rHb over the 72 h time period investigated. Values for flow, SO_2_, and rHb are given in Table 6 and Table 7 as well as Figure 2 and Figure 3. Interestingly, SO_2_ values tend to be lower in those patients that passed away during the course of the study; however, significant differences were only seen at 3, 12, and 18 h post-anastomosis for GM flaps.

## 4. Discussion

Free flap transplantation for soft tissue reconstruction has evolved to the method of choice in defects of significant size and depth, with vulnerable structures such as vessels, tendons, or bones exposed [2,3,4,5,6,7,8,9,10,11,12]. Accordingly, free flap transfer in multimorbid patients has become more important. This single-center retrospective cohort study investigated patient demographics, patient characteristics, comorbidities and medication, perioperative details, microcirculation, and free flap outcome in groups stratified according to patient survival after free flap transfer.

Plastic surgical centers often evaluate free flap transfer according to revision rates, flap outcome, and flap failure. In fact, free flap outcome has significantly increased over time, and has become a highly standardized, safe, and successful means of providing reconstruction of large-sized defects in a broad range of indications [2,3,4,5,6,7,8,9,10,11,12,13,14]. Nevertheless, the patient population requiring microsurgical free flap reconstruction is often challenging, as several comorbidities complicate the procedure and perioperative patient management. As life expectancy at birth among women and men is constantly rising in Europe, patients who need reconstructive surgeries are also older and show numerous comorbidities. Complex microsurgical soft tissue reconstructions should be discussed carefully in this patient group [7,11,15]. This holds especially true for patients treated at specialized large-volume centers, where indications include complicated chronic wounds, wound healing disturbances, and defects after severe trauma, tumor, or infection [11,15].

In addition to postoperative flap outcome, patients’ short- and medium-term survival after free flap surgery should be a factor discussed in preoperative evaluation and the process of informed consent. Surgical and non-surgical alternative therapies can play a role, if the morbidity of the patient is high, and short- to mid-term survival with significant quality of life cannot be guaranteed. Free flap surgery is associated with relatively high surgical costs, and perioperative risks for the patient, caused by the operation itself and the postoperative rehabilitation period [20,21,22]. Hence, both from a health-economic, as well as patient-oriented standpoint, further data are needed regarding potential factors contributing to reduced survival in the short- and mid-term postoperative period after free flap transfer, to aid patients in decision-making and facilitate choosing the best therapeutic option in terms of prolonged quality of life and patient satisfaction. On the other hand, data indicate that free flap surgery reduces hospitalization, as well as the absolute number of operations in patients with complex defects [23,24]. In addition, the morbidity and mortality of amputation, often the alternative to complex extremity reconstruction using microsurgical techniques, are reported to be as high as 40% to 82% after below-the-knee amputation and 40% to 90% (both 5-year mortality rates) after above-the-knee amputation [25]. Therefore, free flap reconstruction plays an invaluable option in this patient cohort.

The data of the presented study show no significant differences with regard to revision surgery in flaps of patients that passed away during the study period and those alive. This is in line with previous data of various study groups stating that free flap surgery can be successfully performed in multimorbid patients, supporting the fact that free flaps themselves provide a safe and effective method of defect reconstruction in the experienced hand [7,15,26]. Yet, a tendency toward a higher rate of revision surgery was seen in those patients that died in due course, with 18% of flaps requiring surgical revision. In addition, flap loss was higher in this patient group, reaching 12%, which is markedly higher compared to flap loss rates reported in the literature, as well as the flap loss rates in patients alive at the end of the study period (3%) [27,28,29]. It is worth noting that patients who died in the short- or mid-term follow-up after surgery had a significantly higher ASA status and suffered significantly more frequently from cardiovascular and metabolic diseases. Consequently, this patient group also significantly more often needed daily anticoagulants as well as antidiabetics. Potentially, this factor could contribute to the increased number of revision surgeries and flap loss in this study group. Previous data have shown that patients with coagulative diseases need a revision surgery after free flap transfer more frequently compared to healthy patients, also corresponding to higher rates of free flap loss [7,30,31,32].

Irrespective of free flap outcome, overall patient survival seems to be negatively affected by the number and type of comorbidities, as well as the preoperative ASA status. These factors should be regarded as risk factors in terms of patient outcome prior to surgery. Importantly, these factors do not only increase the mortality of patients receiving free flap transplantation but are also similarly expected for other operative indications performed by other specialties [33,34]. Apart from comorbidities, differences between groups were also found regarding patient age. Perioperative mortality was increased in older patients, with a mean difference of ~13 years between those patients that died and those alive at the end of the study period. Previous studies have shown successful free flap transplantation in elderly population [35]. Yet, patient survival was not a factor assessed within the postoperative period in these studies and should be included in the preoperative decision-making process in the future [30]. There is a scientific gap with regard to a specific cut-off value for age in free flap surgery. The authors believe that a highly individualized decision based on each individual case is warranted in the elderly. Heidekrueger et al. showed that successful free tissue transfer can also be achieved in a very old population, when comparing patients >80 years and <80 years, despite higher rates of patient comorbidities [35]. On the other hand, Coban et al. reported a case of late flap failure in a 105-year-old patient due to un-autonomization. As the population is growing increasingly old, further studies are needed in this regard [36].

Interestingly, the duration of surgery did not influence outcome in terms of patient survival significantly. Instead, longer hospitalization was observed in those patients that passed away. Longer hospitalization is, however, likely attributable to the increased number of comorbidities, patient age, and preoperative ASA status [37]. No statistically significant differences were observed for defect etiology and defect location between both groups. This supports previous data, favoring free flap transplantation for a variety of indications all over the body [5,6,7,8,9,10]. When taking a closer look at the data, both groups showed nearly the same relative number of tumor resections as defect etiology. Patients that were alive, however, suffered from defects after trauma more frequently, whereas those that had died had suffered from chronic wounds more frequently. Once again, this can be traced back to the study population itself, with younger patients more frequently involved in traumatic accidents, compared to older patients with an increased amount of comorbidity suffering from chronic wounds [38,39].

Analysis of microcirculation in viable free flaps of patients receving GM or LDM flaps showed no significant differences with regard to microcirculatory flow, or rHB when comparing dead and alive patients. Interestingly, SO_2_ values were lower over all measuring timepoints in the deceased group, only reaching statistical significance at 3, 12, and 18 h post-anastomosis for GM flaps, however. Therefore, as was expected, postoperative microcirculation measured using the O2C device cannot be utilized as a predictive value of postoperative patient survival in this patient cohort. Potentially, however, oxygenation of the capillary bed might have been reduced in those patients that passed away due to their comorbidity profile. Future studies using the O2C device must assess the influence of comorbidities that induce micro- and macroangiopathy such as PAD and DMII. Interestingly, measurements for rHb and SO_2_ were overall lower in LDM compared to GM flaps. Similarly, a previous study comparing O2C values in GM and anterolateral thigh (ALT) flaps demonstrated lower values in ALT flaps [40]. The measuring probe in ALT and LDM flaps is placed onto the skin island, i.e., the fasciocutaneous component of the flap, while the probe was sutured directly to the muscle in GM flaps. This likely explains the differences detected, as the probe measures microcirculation to a 7 mm depth.

This study is not free of limitations. The study design as a retrospective analysis of prospectively collected data is a limitation itself and does not generate data of the highest quality. It is pivotal to collect more perioperative data from a larger patient cohort, including further types of free flaps. 

## 5. Conclusions

In conclusion, soft tissue reconstruction utilizing free flap transfer is safe and effective, even in multimorbid patients. Yet, identifying objective predictors for postoperative complications is pivotal to ensure a high success rate of microvascular tissue transfers. Data analysis showed that patient age, comorbidities, preoperative ASA status, and medication significantly impact postoperative patient survival after free flap transfer in the short- and mid-term and must, therefore, be taken into account in preoperative decision-making and informed consent.

## Figures and Tables

**Figure 1 diagnostics-12-02877-f001:**
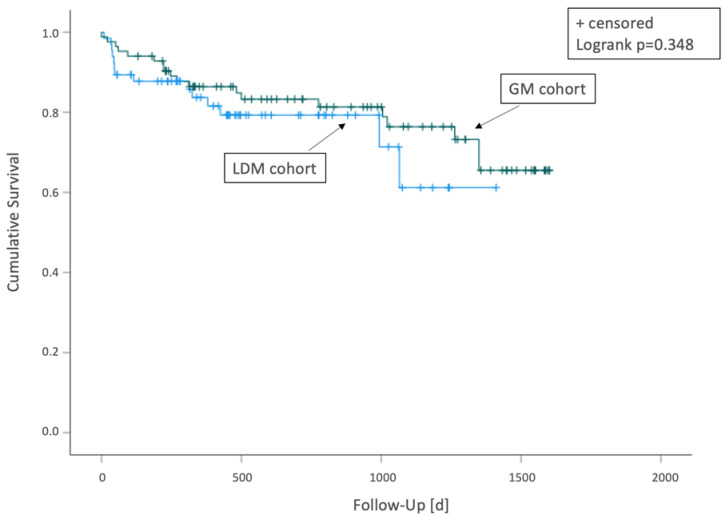
Kaplan–Meier survival analysis for LDM and GM cohort.

**Figure 2 diagnostics-12-02877-f002:**
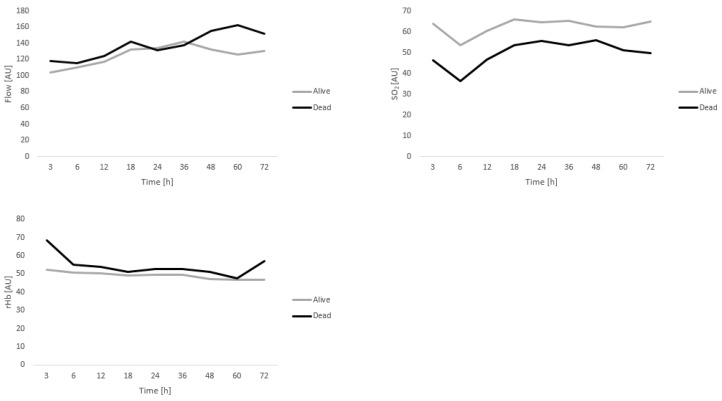
Analysis of microcirculation in viable GM flaps (Alive *n* = 26; Dead *n* = 9). Data are depicted for flow, SO_2_, and rHb over the 72 h time period investigated.

**Figure 3 diagnostics-12-02877-f003:**
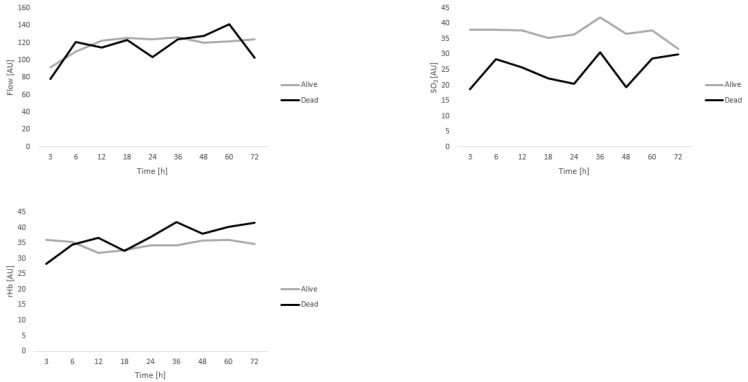
Analysis of microcirculation in viable LDM flaps (Alive *n* = 28; Dead *n* = 4). Data are depicted for flow, SO_2_, and rHb over the 72 h time period investigated.

**Table 1 diagnostics-12-02877-t001:** Patient’s diseases and examples of classification.

Co-Morbidities
Cardiological diseases: arterial hypertension/CAD/arrhythmia/cardiomyopathy
Vascular diseases: PAD/thrombosis
Metabolic diseases: diabetes/osteoporosis/thyroid disease/hyperuricemia
Tumors
Internal diseases: CKI/chronic hematological diseases/COPD/rheumatoid diseases/liver failure
Addiction to substances: nicotine/alcohol/opiate
Neurological diseases: polyneuropathy/ataxia/epilepsies
Musculoskeletal diseases: endoprosthesis/facture
Infectious diseases: hepatitis B

CAD, coronary artery disease; PAD, peripheral artery disease; CKI, chronic kidney disease; COPD, chronic pulmonary disease.

**Table 2 diagnostics-12-02877-t002:** Patient medication and examples of classification.

Medication
Anticoagulant/Antiplatelet: aggregation inhibitors/NOAC/vitamin K antagonists/heparine
Antihypertensives: beta-blockers/RAAS/ Ca^2+^ antagonists/others
Diuretics
Statins
Antidiabetics
Antibiotics/antifungals
Medication for metabolic diseases: L-thyroxin/irenat/allopurinol/colecalciferol
Immunomodulators: cemiplimab/cortisone
Painkillers: (a) NSAID; (b) low-potency opioids; (c) high-potency opioids
Anticonvulsant drugs
Antidepressants
Gastrointestinal-related drugs: pantoprazole/laktulose/pankreatin

NOAC, new oral anticoagulants; RAAS, renin-angiotensin-aldosterone system; NSAID, nonsteroidal anti-inflammatory drug.

**Table 3 diagnostics-12-02877-t003:** Demographic patient information and detailed operative information. Means and frequencies are based on the number of free flaps performed.

Variable	Alive		Dead		*p*-Value
Age (Y)	58.29	SD 17.91	71.39	11.13	<0.001
Gender					
male	67	56.80%	18	54.50%	
female	51	43.20%	15	45.50%	0.819
Type of flap					
LDM	53	44.90%	14	42.40%	
GM	65	55.10%	19	57.60%	0.799
Time of Surgery (min)	327.14	SD 99.25	332.48	95.17	0.803
Flap Ischemia (min)	48.47	SD 16.13	49.69	15.04	0.534
Hospitalization (d)	29.66	SD 26.97	36.88	31.26	0.046
Revision					
no	105	89.00%	27	81.80%	
yes	13	11.00%	6	18.20%	0.371
Flap loss					
no	114	96.60%	29	87.90%	
yes	4	3.39%	4	12.12%	0.069
Defect etiology					
Tumor	49	41.50%	14	42.40%	
Trauma	38	32.20%	7	21.20%	
Chronic Wound	31	26.30%	12	36.40%	0.373
Defect location					
Head and Neck	30	25.40%	9	27.30%	
Trunk	8	6.80%	3	9.10%	
Upper extremity	18	15.30%	2	6.10%	
Lower extremity	62	52.50%	19	57.60%	0.574

Y, years; LDM, latissimus dorsi flap; GM, gracilis muscle flap; min, minutes; d, days; SD, standard deviation.

**Table 4 diagnostics-12-02877-t004:** Patients’ comorbidities according to the group: alive vs. dead. Means and frequencies are based on the number of free flaps performed.

Variable	Alive		Dead		*p*-Value
Cardiovascular					
no	75	63.60%	11	33.30%	
yes	43	36.40%	22	66.70%	0.002
Vasculopathy					
no	89	75.40%	24	72.70%	
yes	29	24.60%	9	27.30%	0.752
Metabolic Disease					
no	78	66.10%	15	45.50%	
yes	40	33.90%	18	54.50%	0.031
Tumor					
no	84	71.20%	18	54.50%	
yes	34	28.80%	15	45.50%	0.071
other internal medicine disease					
no	74	62.70%	16	38.50%	
yes	44	37.30%	17	51.50%	0.141
Neurologic					
no	92	78.00%	28	84.80%	
yes	26	22.00%	5	15.20%	0.387
Musculoskeletal					
no	85	72.00%	26	78.80%	
yes	33	28.00%	7	21.20%	0.437
Infectious					
no	108	91.50%	31	93.90%	
yes	10	8.50%	2	6.10%	1
Addictives					
no	93	78.80%	26	78.80%	
yes	25	21.20%	7	21.20%	0.997
ASA Score					
1	3	2.50%	0	0.00%	
2	42	35.60%	2	6.10%	
3	69	58.50%	28	84.80%	
4	4	3.40%	3	9.10%	0.004

ASA, American Society of Anesthesiologists (physical status classification system).

**Table 5 diagnostics-12-02877-t005:** Patients’ medication according to the group: alive vs. dead. Means and frequencies are based on the number of free flaps performed.

Variable	Alive		Dead		*p*-Value
Anticoagulant/Antiplatelet					
no	49	41.50%	6	18.20%	
yes	69	58.50%	27	81.80%	0.014
Antihypertensive					
no	71	60.20%	15	45.50%	
yes	47	39.80%	18	54.50%	0.131
Diuretics					
no	89	75.40%	20	60.60%	
yes	29	24.60%	13	39.40%	0.093
Statins					
no	92	78.00%	23	69.70%	
yes	26	22.00%	10	30.30%	0.324
Antidiabetics					
no	103	87.30%	23	69.70%	
yes	15	12.70%	10	30.30%	0.016
Antibiotics/Antimycotics					
no	72	61.00%	16	48.50%	
yes	46	39.00%	17	51.50%	0.197
Metabolism					
no	80	67.80%	12	36.40%	
yes	38	32.20%	21	63.60%	0.001
Immunomodulators					
no	107	90.70%	29	87.90%	
yes	11	9.30%	4	12.10%	0.742
NSAID					
no	60	50.80%	19	57.60%	
yes	58	49.20%	14	42.40%	0.494
Low-potency opioid					
no	99	83.90%	30	90.90%	
yes	19	16.10%	3	9.10%	0.410
High-potency opioid					
no	106	89.80%	29	87.90%	
yes	12	10.20%	4	12.10%	0.752
Anticonvulsant					
no	96	81.40%	30	90.90%	
yes	22	18.60%	3	9.10%	0.192
Antidepressant					
no	104	88.10%	27	81.80%	
yes	14	11.90%	6	18.20%	0.385

NSAID, nonsteroidal anti-inflammatory drug

**Table 6 diagnostics-12-02877-t006:** Detailed analysis of the evolution of microcirculation in viable GM flaps over a period of 72 h post-anastomosis (Alive *n* = 26; Dead *n* = 9). Mean values are given with Standard Deviation (SD).

Time (h)		Flow	SD	*p*-Value	SO_2_	SD	*p*-Value	rHb	SD	*p*-Value
3	Alive	103.73	38.67		63.81	14.84		52.5	12.57	
	Dead	118	56.30	0.45	46.43	11.73	0.01	68.43	18.25	0.01
6	Alive	110.45	40.14		53.43	18.48		50.67	13.341	
	Dead	115.43	45.91	0.79	36.43	25.30	0.07	55.14	16.96	0.48
12	Alive	117.08	39.25		60.29	13.21		50.17	12.98	
	Dead	124	493.5	0.72	46.67	19.50	0.05	53.83	21.61	0.60
18	Alive	131.96	40.26		65.87	14.03		49.04	13.46	
	Dead	142.17	49.01	0.6	53.67	13.78	0.01	51.17	16.46	0.74
24	Alive	134.15	37.68		64.65	19.14		49.4	13.10	
	Dead	131.25	47.01	0.86	55.75	13.80	0.24	52.75	12.37	0.54
36	Alive	141.95	34.62		65.36	13.57		49.63	11.76	
	Dead	138	43.93	0.78	53.5	15.85	0.05	52.62	10.66	0.53
48	Alive	131.94	23.82		62.44	12.01		47.28	12.87	
	Dead	155	47.15	0.14	55.8	16.39	0.32	51.2	11.73	0.55
60	Alive	126.31	32.51		62.19	11.61		46.69	13.25	
	Dead	162.75	48.67	0.08	51	9.59	0.09	47.5	7.50	0.90
72	Alive	130.14	25.78		65	17.04		47	16.06	
	Dead	151.75	56.35	0.40	49.75	15.90	0.18	57.25	9.14	0.28

**Table 7 diagnostics-12-02877-t007:** Detailed analysis of the evolution of microcirculation in viable LDM flaps over a period of 72 h post-anastomosis (Alive *n* = 28; Dead *n* = 4). Mean values are given with Standard Deviation (SD).

Time (h)		Flow	SD	*p*-Value	SO_2_	SD	*p*-Value	rHb	SD	*p*-Value
3	Alive	91.55	37.05		38	24.94		36.1	10.73	
	Dead	78.83	21.13	0.56	18.67	18.58	0.21	28.33	1.53	0.23
6	Alive	110	42.51		37.92	18,.42		35.5	9.62	
	Dead	120.75	53.74	0.65	28.5	14.15	0.34	34.5	10.15	0.85
12	Alive	122.69	54.92		37.73	20.65		31.92	10.84	
	Dead	114.25	44.57	0.77	25.75	12.12	0.27	36.75	13.30	0.43
18	Alive	125.88	52.90		35.27	21.40		32.77	8.39	
	Dead	123	40.68	0.92	22.25	10.90	0.25	32.5	10.78	0.95
24	Alive	123.8	42.02		36.52	20.49		34.32	7.12	
	Dead	103.75	28.42	0.37	20.5	10.78	0.14	37	17.18	0.58
36	Alive	126.30	36.14		41.91	20.12		34.39	7.94	
	Dead	124	22.04	0.90	30.75	9.29	0.29	41.75	7.63	0.1
48	Alive	119.75	25.14		36.58	19.47		35.92	12.11	
	Dead	128	35.51	0.61	19.33	13.50	0.15	38	8.54	0.78
60	Alive	121.52	39.28		37.7	18.83		35.96	10.78	
	Dead	141.33	69.02	0.46	28.67	8.62	0.43	40.33	11.68	0.52
72	Alive	123.63	57.15		31.79	17.02		34.74	8.39	
	Dead	103	1.41	0.62	30	1.41	0.89	41.5	10.61	0.30

## Data Availability

Not applicable.

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
