# Peer review of "Analysis of Factors Determining Patient Survival after Receiving Free-Flap Reconstruction at a Single Center—A Retrospective Cohort Study"

_diagnostics, 2022, doi:10.3390/diagnostics12112877_

Round 1
Reviewer 1 Report
Dear Authors,
Thank you very much for submitting this interesting paper. I have no doubt that our colleagues will find this work to be fascinating and useful to read.
I have got just a curiosity.
Line 84 “All patients receiving free GM or LDM flaps” why did you decide to analyse only patients treated with these 2 muscular flaps?
Thank you.
Author Response
We thank the reviewer for this interesting question. These two flaps were chosen as they are considered as standard of care in our department, with low morbidity and high safety.
Reviewer 2 Report
The authors made a retrospective analysis of free flap surgery patients. The methodology and statistical analysis, data presentation and discussion is greatly appreciated. However some issues remain unanswered as follows:
1-Main question adressed by the research, evaluate the outcome for multimorbid patients who underwent microsurgical soft tissue reconstruction. and to identify potential risk factors that may increase patient death.It is well known that postoperative complications increases with increased ASA scores. ASA 3-4 scores were higher in old age group. They confirmed this scientific finding in the study.Additionally they also showed that
-duration of surgery did not influence outcome in terms of patient survival significantly
-longer hospitalization was observed in the dead group
- a tendency toward a higher rate of revision surgery was seen in those patients that died
2-In the literature there is scientific gap regarding to answer to the following question;
“ Is there any cut-off limit for age in free flap surgery? Is there any scientific explanation regarding to organ functions limiting longer surgeries at this age?”
I present an example from the literature as given below;
“ Free Radial Forearm Flap Failure due to Un-Autonomization in a 105-Year Old Patient.YK Coban, K Bekircan, O Ocuk, OG Dinc, N Karadag.Clin Surg. 2017; 2 1608.”
The authors does not address to this in the study.
3- Free flap surgery needs near normal biochemical and hematological parameters. As comorbidity and older age corelates with altered medical parameters, longer surgeries are expected to be more complicated. In this study, this finding is confirmed. This confirmation is important for reinforcement of current knowledge in the subject area.
4-Serum albumin and fibrinogen levels are very important in wound healing period. So, I advice the authors to include these parameters into account when conducting free flap analysis in further studies .
5-Conclusions are greatly appreciated.
6-My above mentioned reference may be added to the list with a small expansion of the discussion section.
7-I find all tables and figures useful.
Author Response
We thank the reviewer to raise these points and highly appreciate these comments.
The authors agree that there is a scientific gap with regard to a specific cut-off value for age in free flap surgery. However, the authors believe that a highly individualized decision based on each individual case needs to be made in the elderly. Several publications have demonstrated the feasibility of free flap transfer in old age. Heidekrueger et al. (2016) showed that successful free tissue transfer can also be achieved in a very old population, when comparing patients >80 years and <80 years, despite higher rates of patient comorbidities. However, as the population is increasingly aging, further studies such as the one proposed by the reviewer (Conan et al.) are needed to investigate flap surgery in even older patients. We have included this within our discussion section. In addition, we agree with the reviewer that more details on wound healing would help to interpret the results. We will include more parameters like serum albumin and fibrinogen levels in our future studies.
Reviewer 3 Report
Well-written text. Interesting method chosen for patients. A large research patient group. Nevertheless, I am missing some data. Although the authors write that locations the defect did not matter to the reader, it is interesting. I think that the combination of the presented data with the length of the surgical procedure (vs the location and type of the flap) would be of research significance.
Author Response
We thank the reviewer for the supportive comment and understand the criticism. However, a detailed analysis of the length of the surgical procedure based on the location and type of flap was not aim of the study. Both groups showed similar distribution of flap types, defect location and surgical time. Even if there were differences in the operative time between flap types and defect location, these would not impact the outcome of our study as the distribution was comparable between alive and dead patients.
For courtesy of the reviewer we provide the operative times here, without including them in the manuscript: Mean operative time was 338 min for head and neck, 425 min for trunk, 300 min for upper and 316 for lower extremity defects. For LDM flaps mean operation time was 364 min, while it was 298 min for GM flaps.
Reviewer 4 Report
The reference list of the manuscript contains 39 titles, and is without inappropriate self-citations. One references is elder than 20 years. The manuscript is clear, with a high rate of clinical significance. The manuscript present scientific resound and the design appropriate to test the hypothesis. The methods are clear described, with sufficient details to permit another researcher to reproduce the results. All aspects regarding the figures/tables/images are appropriate, and they are easy to interpret and understand. The presentation and the analyzed date are written in proper way. The presentation of the results are at high standard, with appropriate statistics. The results offer a development in the present knowledge, are significant, and are suitable interpreted.
If have only some comments
LL63-65:
This is an important aspect of free flap surgery. It is significant to highlight the problems of hemodynamics, listing the need for blood products and also catecholamines. For example: Identifying perioperative volume-related risk factors in head and neck surgeries with free flap reconstructions - An investigation with focus on the influence of red blood cell concentrates and noradrenaline use. Grill FD, Wasmaier M, Mücke T, Ritschl LM, Wolff KD, Schneider G, Loeffelbein DJ, Kadera V. J Craniomaxillofac Surg. 2020 Jan;48(1):67-74. doi: 10.1016/j.jcms.2019.12.001. Epub 2019 Dec 16. PMID: 31874805
LL 166: Please add standard deviation
LL 206 and 210: I suggest adding the number of cases (dead/alive) of the analyzed flap types in the table header. That simplifies reading.
LL 267-276:
Certainly, the ASA score is a viable parameter that allows orienting classification of patient comprehension, but its discriminatory power between ASA 2 and 3 and 4 should be considered. This idea should perhaps be added to your discussion. Did you assessed Charlon´s comorbidity index (CCI)?
LL 318-324:
Both types of flaps play a major role in reconstructive surgery. From my point of view, there are limitations in direct comparability. What about flap volume, donor site infection? Does flap type selection affect morbidity itself?
Author Response
We thank the reviewer for the important comments. We have included the proposed reference within the introduction section. We have added the standard deviation in Table 1. We have added the number of viable flaps measured using the O2C device in the table headings. For our analysis, we aimed to work in a non‑biased manner. The ASA- score is determined by the colleagues of the anesthesiology department and therefore it is not dependent on our evaluation. We did not asses the CCI. Certainly, flap type selection can affect patients morbidity. However, in both groups investigated (dead and alive) in this study, there was no significant difference in the flap type chosen for reconstruction. So as far as our results go, choice of flap did not affect mortality of patients.